# Probing the Boundary between Classical and Quantum Mechanics by Analyzing the Energy Dependence of Single-Electron Scattering Events at the Nanoscale

**DOI:** 10.3390/nano13060971

**Published:** 2023-03-08

**Authors:** Christian Kisielowski, Petra Specht, Stig Helveg, Fu-Rong Chen, Bert Freitag, Joerg Jinschek, Dirk Van Dyck

**Affiliations:** 1The Molecular Foundry, Lawrence Berkeley National Laboratory, One Cyclotron Rd., Berkeley, CA 94720, USA; 2Department of Materials Science and Engineering, University of California Berkeley, Berkeley, CA 94720, USA; 3Center for Visualizing Catalytic Processes (VISION), Department of Physics, Technical University of Denmark, 2800 Kongens Lyngby, Denmark; 4Department of Materials Science and Engineering, City University of Hong Kong, Kowloon Tong, Hong Kong SAR, China; 5Thermo Fisher Scientific, Achtseweg Noord 5, 5651 GG Eindhoven, The Netherlands; 6National Centre for Nano Fabrication and Characterization (DTU Nanolab), Technical University of Denmark, 2800 Kongens Lyngby, Denmark; 7EMAT, University of Antwerp, 2020 Antwerp, Belgium

**Keywords:** electron beam–sample interactions, functional behavior, inelastic scattering, time-dependent Schrödinger equation, self-interference, coherence, Heisenberg’s uncertainty principle, wave packets

## Abstract

The relation between the energy-dependent particle and wave descriptions of electron–matter interactions on the nanoscale was analyzed by measuring the delocalization of an evanescent field from energy-filtered amplitude images of sample/vacuum interfaces with a special aberration-corrected electron microscope. The spatial field extension coincided with the energy-dependent self-coherence length of propagating wave packets that obeyed the time-dependent Schrödinger equation, and underwent a Goos–Hänchen shift. The findings support the view that wave packets are created by self-interferences during coherent–inelastic Coulomb interactions with a decoherence phase close to Δ*φ* = 0.5 rad. Due to a strictly reciprocal dependence on energy, the wave packets shrink below atomic dimensions for electron energy losses beyond 1000 eV, and thus appear particle-like. Consequently, our observations inevitably include pulse-like wave propagations that stimulate structural dynamics in nanomaterials at any electron energy loss, which can be exploited to unravel time-dependent structure–function relationships on the nanoscale.

## 1. Introduction

Aberration-corrected transmission electron microscopy (TEM) is the outstanding method to analyze the structure and chemical composition of nanomaterials with single-atom sensitivity across a large length scale ⪝1 micrometer with sub-Ångstrom resolution [1,2]. Moreover, data modeling based on the multislice (MS) approximation [3] has been successfully implemented to accurately match and predict the numerous experimental TEM observations of atomically resolved, static nanostructures if frozen phonon excitations are included in the coherent–elastic calculations [4,5]. However, new challenges emerge as TEM technologies rapidly evolve to advance a high temporal resolution of <1 ps at a high spatial resolution of <1 Å [6,7,8] because beam–sample interactions [9] and quantum effects [10,11,12] become increasingly relevant in the quest to promote scientific insight into the structural dynamics of nanomaterials.

Typically, the Copenhagen convention of quantum mechanics [13] describes the boundary between the quantum world, where wave functions are indefinite in space and time, and the classical world of particles that are perceived as definite. Heisenberg’s uncertainty principle defines this boundary by:Δ*t* Δ*E* ⩾ *ℏ* /2(1)
where Δ*t* is a time difference, Δ*E* is an energy change, and *ℏ* is the reduced Planck constant. However, its relationship to scattering processes and the wave–particle (or field–particle) duality remains surprisingly controversial because of limited experimental access [14]. Nowadays, this duality can be detected in every interference experiment at the ultimate detection limit, where isolated electron-scattering events and self-interferences are directly observable in chromatic (C_c_) and spherical (C_s_) aberration-corrected electron microscopes [12]. In this context, it is desirable to describe Coulomb-scattering processes of charged electrons [15] using Schrödinger’s *time-dependent* equation:(2)$$-\frac{\hbar^{2}}{2m}\nabla^{2}\psi \left(r,t\right)+V\left(r\right)\psi \left(r,t\right)=i\hbar \frac{d\psi \left(r,t\right)}{dt}$$
which can describe coherent–inelastic Coulomb interactions [15] instead of the widely used static approximation, where the right-hand side of Equation (2) is set to zero. Here, *ψ*(*r*,*t*) is the electron wave function in space and time, *V*(*r*) is the electrostatic scattering potential, and *m* is the electron mass.

In this paper, we show that aberration-corrected, energy-dependent electron scattering in surface proximity is accounted for by Equation (2) rather than the *time-independent* Schrödinger equation that forms the basis for the above-mentioned MS approximation. Our experiments on the chromatic and spherical aberration-corrected TEAM I microscope were performed in the limit of single-electron self-interferences in the absence of contrast (=information) delocalization due to finite lens aberrations [12,16]. Surprisingly, we found that our observed delocalization of information in recorded amplitude images was dependent only on the energy loss during the interaction of wave functions and tracks in the Heisenberg relation (1), while the wave character of the scattered electrons was always maintained by the formation of propagating wave packets (pulses) with energy-dependent widths. As an example of the impact of our results on science at the nanoscale, we point out that it recently became possible to estimate and manipulate the critical accumulated electron dose for the irradiation-induced decay of crystal structures by better understanding time-dependent phonon scattering in both radiation-hard and radiation-soft materials [17].

## 2. Materials and Methods

### 2.1. Measurement Principle

In order to investigate how electron scattering contributes to Equation (2), we extracted constants that were experimentally accessible. This was possible because it is common practice to obtain solutions of Schrödinger’s *time-dependent* equation through a separation of the temporal (*t*) and spatial variables (*r*) as *ψ*(*r*,*t*) = *ψ*(*r*) *φ*(*t*), which yields:(3)$$-\frac{\hbar^{2}}{2m}\frac{1}{\psi \left(r\right)}\nabla^{2}\psi \left(r\right)+V\left(r\right)=E=i\hbar \frac{1}{\varphi \left(t\right)}\frac{d\varphi \left(t\right)}{dt}$$

Thus, the *time-independent* part of the solution obeys the differential equation:(4)$$\nabla^{2}\psi \left(r\right)=-\frac{2m}{\hbar^{2}}\left(E-V\left(r\right)\right)\psi \left(r\right)=-k^{2}\psi \left(r\right)$$

For spatial solutions of Equation (3), such as a wave function trapped in a potential barrier, there is the well-known exponential penetration of intensity into the potential wall, which is exploited in scanning tunneling microscopy (STM) [18]:(5)$$\psi \left(x\right)=Be^{-kx}$$

It creates a stationary evanescent field that decays over the tunneling distance *l_t_* = 1/*k*
(6)$$l_{t}\left(\Delta E\right)=\frac{\hbar }{\sqrt{2m\left(E-V\left(r\right)\right)}}$$
as shown in Figure 1a. It can be seen that the tunneling distance *l_t_* depends on the inverse square root of energy and originates from the second derivative (curvature) of Schrödinger’s *time-independent* equation.

Similarly, the *time-dependent* part of Equation (3) is given by:(7)$$\frac{d}{dt}\varphi \left(t\right)=\frac{E}{i\hbar }\varphi \left(t\right)$$
which solves as:(8)$$\varphi \left(t\right)=\varphi \left(0\right)e^{\frac{-iEt}{\hbar }}$$
and oscillates in time with the characteristic time constant *t_e_* = *ℏ*/*E* that can be transformed into a length scale by multiplication with the speed of light *c*:(9)$$l_{e}\left(E\right)=\frac{\hbar x}{tE}=\frac{\hbar c}{E}$$

Unlike Equation (6), the characteristic length *l_e_* of Equation (9) depends on the inverse energy because it originates from the first derivative of Schrödinger’s *time-dependent* equation.

From an optical perspective, grazing incident angles of wave functions that occur at sample/vacuum interfaces satisfy the total internal reflection conditions. In turn, these scattering conditions create a transversal evanescent field within a coherently illuminated area that decays exponentially into the vacuum. This process is schematized in Figure 1b and explained by the Goos–Hänchen shift (GHS) *D* [19]. The effect is exploited by photon STM [20] or fluorescence techniques [21]. During observations with an electron microscope, the total reflection of electron waves always occurs at the side faces of the investigated samples (Figure 1b). Moreover, the equivalence of the effect for quantum mechanical electron wave functions (particle fields) and optical wave functions (electromagnetic fields) was explicitly pointed out some time ago [22]. It allows estimating the extension of the created evanescent field, independent of the field or particle nature of the incident irradiation by:(10)$$l_{e}\left(\Delta E\right)≅D=\frac{\lambda }{\pi }\frac{\sin\left(\Phi \right)}{\sqrt{\sin{\left(\Phi \right)}^{2}-n^{2}}}⩾\frac{A\hbar c}{\Delta E}$$

In Equation (10), we apply the Heisenberg relation (1) with wavelengths *λ* = Δ*λ* = *c* Δ*t* and estimate the constant *A* ≂ 1 by using shallow incident angles ~1^0^ together with the relative refraction index *n* = *n*(sample)*/n*(vacuum) close to unity [23] (here: *n* = 0.99). This choice highlights the similarity of Equations (9) and (10), which are derived from rather different points of view, as well as their direct relation to Heisenberg’s uncertainty principle. Thereby, it is understood how both the static and the dynamic part of Schrödinger’s *time-dependent* Equation (3) can contribute to the formation of an evanescent field at sample/vacuum interfaces in electron-irradiated samples. Now, the task at hand is to identify the dominant mechanism, which can be distinguished by measuring the energy dependence of the penetration depth *l_t_*~1/√*E* or *l_e_*~1/*E* that we investigate here.

### 2.2. Aberration-Corrected, Energy-Filtered Amplitude Imaging with Single-Electron Detection

The very special TEAM I microscope at the Molecular Foundry was developed by the Department of Energy to promote deep sub-Ångstrom resolution by advancing C_c_ and C_s_ aberration corrections and camera performance [24]. Recently, it was demonstrated that it allows modulating beam–sample interactions by dose fractionation with ultra-low dose rates that routinely capture isolated scattering events [12]. Briefly, the microscope is operated in a Nelsonian illumination mode that is suitably generated by a monochromator/gun assembly, with chromatic and spherical aberration correction, and a direct electron detector (K2 camera). It creates a pencil-like, highly coherent electron beam (Δ*E* = 0.1 eV) that matches *a* ~10^7^ Å^2^ large field of view, which is captured on the direct electron detector. This operating mode allows the merging of established practices of cryogenic electron microscopy (cryo-EM) with high-resolution transmission electron microscopy (HRTEM). In particular, investigations of radiation-sensitive soft matter become feasible on a level that can reach beyond the successful efforts of Cryo-EM by including time resolution [8,12].

For the current experiments, a K3 camera was mounted behind the Gatan Image Filter (GIF). Electrons accelerated by 300 kV have a de Broglie wavelength of 2 pm and travel at 78 percent of the speed of light. Their wavelength increases by Δ*λ* if energy losses Δ*E* occur during their interactions with solids in a vacuum. Energy dependences can be captured in real space images with single-electron sensitivity by performing energy-filtered transmission electron microscopy (EFTEM) [1]. In this case, the energy resolution was set by the energy spread of the monochromator Δ*E_mc_* and the energy-selecting slit Δ*E_s_* of the GIF. We used gold nanocrystals on amorphous carbon to calibrate two microscope settings for imaging with electron beams of energy spread Δ*E_mc_* = 0.6 eV and Δ*E_mc_* = 0.1 eV at full width half maximum together with slit widths of Δ*E_s_* = 1 or 2 eV and Δ*E_s_* = 0.3 eV, respectively. These conditions differ in terms of irradiated areas and beam coherence, where either the C_2_ aperture determines the field of view or the aforementioned Nelsonian illumination scheme (Figure 2a). Flat-field corrections were performed by the standard GIF alignment procedure, which minimizes chromatic fluctuations across the field of view that become noticeable for energy losses below ~1.5 eV. Phonon losses occur below this energy threshold. The presence of a C_c_ corrector ensures that all coherent–inelastically scattered electrons with energy losses below ~600 eV are focused in the same imaging plane [16]. We analyzed the EFTEM images recorded with energy losses between 1 and 80 eV because the aforementioned GIF distortions imposed a low energy limit and electron counts dropped below reasonable values for losses approaching 100 eV.

### 2.3. Materials and Methods

The free-standing GaN samples in our experiments were produced by a focused ion beam (FIB) milling process using a low-dislocation density GaN substrate from Furukawa Electric Co, which was of a Wurtzite structure. They were prepared in an a-plane [100] sample orientation with a Helios FIB from Thermo-Fischer. Any residual surface damage was removed by cleaning with low-energy (500 V) argon ions in a Nanomill from Fishione Inc. The process yielded plane parallel samples, together with a graded wedge at the c-plane, and an abrupt m-plane sample/vacuum interface, as shown in Figure 2b. We determined a sample thickness of 290 nm by calculating that Pendellösung oscillations were of 48 nm periodicity, which formed the visible contrast oscillations along the wedge. The sample cleaning process created atomically stepped surfaces (Figure 2b). Our data were taken from the nonpolar m-plane of GaN, recorded with a projected pixel size of 0.0752 nm^2^/pixel, and recorded at times between 0.1 and 100 s. Each image frame consisted of 3456 × 3456 pixels.

For each energy setting, we recorded the incident electron distribution (I) by capturing an empty image frame and the scattered intensity (I_s_) by shifting the sample into the field of view without changing any other recording parameter. The resulting difference images (I_s_-I) were used to quantitatively study the spatial intensity distribution at the GaN/vacuum edge (Figure 3a). Line profiles across the Ga/N interface revealed that single-electron scattering events were detected with outstanding signal-to-noise (S/N) ratio, which is why single image pixel profiles were entirely dominated by telegraph noise (Figure 3b). For noise reduction, image series comprising up to 100 images were recorded, aligned, and averaged. To produce a final profile, we additionally averaged 500–1000 pixels parallel to the abrupt interface.

At solid/vacuum interfaces, Fresnel phase-contrast fringes usually dominate the intensity distribution in TEM images [1] (Figure 4a) and appear with high spatial frequencies on the length scale of 100 nm in the line profiles of Figure 4b. Experimentally, they are suppressed by recording amplitude images at a defocus value close to Δ*f* = 0 nm if all other lens aberrations are negligible. The recording of amplitude images with the TEAM I microscope bears the additional advantage that the amplitude contrast transfer function transmits all low spatial frequencies and they are focused in a constant image plane [1,16]. Figure 4b also shows that an evanescent intensity decay occurred over tenths of a nanometer from the interface into the vacuum region. Based on the implemented approximations, state-of-the-art MS calculations of amplitude images were insufficient to simulate evanescent intensity (Figure 4a, inset) [5]. At present, the MS calculations make use of Huygen’s principle for coherent–elastic interferences of wave functions [3], but do not include coherent–inelastic Coulomb scattering processes [12,17]. Certainly, the literature which exists to describe electron excitation in solids, including imaging in the optical frequency range [25] and aloof spectroscopy [26,27,28], is extensive. However, to our knowledge, energy-filtered amplitude images of evanescent fields using single-electron interferences and a C_c_- and C_s_-corrected microscope have not been reported so far.

## 3. Results

We report in Figure 5 that the amplitude image intensity of evanescent fields decayed in a strictly exponential manner from the interface into the vacuum. Moreover, the decay was accelerated by increasing the electron energy losses Δ*E* from 0.9 eV to 40 eV and remained unaffected by our two choices of energy spread of the monochromator (Figure 5a,b). Similarly, Figure 6a shows two averaged data sets of single measurements that were executed with the same imaging parameters, but with recording times that differ by a factor of 100. They prove that recording times in the range of seconds did not affect the results either. The electron count values show that single-electron interferences were captured, and a 0.2% accurate value of the penetration depth *x_i_* could be extracted from each measurement by a least square fit procedure to the function:(11)$$I\left(xi\right)=I\left(0\right)e^{\frac{-x}{xi}}$$

It is understood that *I*(0) captures the intensity of the electron energy loss spectrum (EELS) at the chosen energy loss [1]. Most importantly, however, the exponential decay constants measured at various energy losses are plotted in Figure 6b, which indisputably prove that the energy loss dependence of the penetration depth followed the rule *x_i_* ≂ *l_e_* ~ 1/*E*. We now make the ansatz that this exponential decay with relates to the formation of an evanescent field by the GHS being projected into the image, analogously to the light optical phenomenon [22]. It is consistent with a description of electron scattering by Schrödinger’s *time-dependent* equation. In this model, the fitting parameter *1/a* in Figure 6b can be related to the elemental constant *ℏc* for wave propagations in a vacuum at the speed of light when noting that the accelerated electrons travel at the speed *v* < *c,* and 1/a = *ℏv* = 106 eVnm < 197 eVnm = *ℏc* ((Equation (9)). This close agreement allows the interpretation of the penetration distance as a natural “delocalization distance” imposed by Heisenberg’s uncertainty principle, and allows estimating electron scattering cross-sections by the self-coherence length *l_s_* at any energy loss. The impact of this surprisingly clear-cut result in favor of the *time-dependent* coherent–inelastic Coulomb interaction of wave functions is substantial because it calls for expanding current MS calculations to include the time domain if one aims to describe dynamic behavior.

## 4. Discussion

It is convenient to discuss coherent–inelastic electron scattering in terms of coherent wave interferences. Coherence is commonly ascribed to any electron ensemble using a temporal (longitudinal) coherence length *l_ct_* and a spatial (transversal) coherence length *l_cd_* that are given by *l_ct_
*= *λ*^2^/Δ*λ* and *l_cd_* = *λ*/2πΔ*θ*, respectively [1]. *λ* is the de Broglie wavelength, and Δ*λ* and Δ*θ* are uncertainties in the energy spread and the angular spread of the electron source, respectively. In electron microscopy, this ensemble limited coherence length is typically included as a partial coherence in damping parameters to the contrast transfer function of the microscopes’ objective lens [1]. Thereby, any contribution from isolated electron scattering events that surely dominate in our experiments with single-electron detection is ignored. Moreover, electron scattering is commonly perceived as either coherent-elastic or incoherent-inelastic [1]. However, there is growing evidence that Coulomb interactions in solids are always inelastic [15,17], and definitions of coherence or decoherence remain ambiguous. Consequently, the description of electron scattering must be considered carefully with the aim to explain already-diverging interpretations (for example, compare [29,30] or [9,31]) that often relate to the excitation of structural dynamics by the electron beam itself.

The superposition principle is one of the strongest principles in quantum mechanics, with a remarkable link to self-interferences and decoherence that was recently reported [12]. It can be used to express the formation of wave packets by assuming the coexistence of incident wave functions with scattered wave functions of wavelength differences Δ*λ* that self-interfere during Coulomb interactions with energy loss Δ*E*. In this case, a self-coherence length *l_s_* can be calculated from a phase (path) difference Δ*φ* of superimposing partial waves by:(12)$$l_{S}=\frac{\Delta \varphi }{\frac{2\pi }{\lambda }-\frac{2\pi }{\lambda \;+\;\Delta \lambda }}$$

For Δ*φ* = 1 rad, the description reproduces the established longitudinal coherence criterion *l_ct_* = *λ*^2^/Δ*λ* for photon ensembles in optics. Here, for the first time, this is now applied to electron waves. The significance of choosing a phase shift Δ*φ* = 0.5 rad for single-electron waves became evident when the calculated self-coherence length *l_s_* was divided by the speed of light to give a self-coherence time *t_s_* = *l_s_*/*c*, as plotted versus the energy loss (Δ*E^−1^*) in Figure 7. Since the slope of the straight line Δ*t_s_* Δ*E* was given by *h*/4π = *ℏ*/2, where *h* was the Planck constant, the depicted linear relation described Heisenberg’s uncertainty principle Δ*t* Δ*E* ≥ *ℏ*/2 for self-interferences, which marked the boundary between classical mechanics (white) and quantum mechanics (pink). In this picture, the self-coherence length of scattered electrons rapidly shrink with increasing energy losses associated with phonon, plasmon, or core excitations in the probed material. Continuously narrowing with the increasing strength of the Coulomb interaction, the electron wave packets seemingly collapsed once *l_s_* reached atomic dimensions at energy losses around 1000 eV. However, even in this case, the Coulomb scattering remained coherent and inelastic, i.e. the short self-coherence length made the wave functions appear particle-like.

This graph suitably compares our measurements to the self-coherence lengths *l_s_
*(Equation (12)), the tunneling distances *l_t_* (Equation (6)), and the extension of evanescent fields *l_e_* created by wave packets traveling at the speed of light *c* or at 54% of the speed of light (Equation (9)). Electrons at 300 kV propagate in a vacuum at 78% of the speed of light, which is altered if the propagation occurs at an interface, and an evanescent field is formed. The best description of our imaging experiments was achieved by assuming that wave packets travel at 54% of the speed of light if interacting with GaN surfaces to create an evanescent field via a GHS (Equation (10)). Given the uncertainties in estimating the constant *A* of Equation (10), it is astounding how well the evanescent field tracked the self-interference of coherent-inelastically scattered electrons at the Heisenberg limit. Reliably, the penetration depth *x_i_* exceeded the self-coherence length *l_s_,* as is required by the uncertainty principle. Moreover, the results are fully compatible with aloof experiments [26,27,28] that also exploited the spatial extension of the evanescent field, as indicated in Figure 7. In our analysis, the electron energy loss spectrum (EELS) is included in the prefactor *I*(0) of the exponential decay function (11).

Therefore, coherent–inelastic interactions likely generate pulse-like wave-packages at any energy loss. Conceptually, the self-interference of *ψ*(*r*,*t*) is treated as a probability amplitude for interactions with energy exchange Δ*E* to occur at *r_o_* and *t_o_*. They must be considered to explain electron scattering beyond static models and allow controlling and exploiting beam-induced structural dynamics. Alternatively, a virtual particle exchange between coexisting wave functions would have to be discussed, which mixes particle and field pictures. Experimentally, it has already been demonstrated that the self-coherence length can be used to define electron scattering cross-sections at any energy [17]. In turn, this allows showing that inelastic phonon scattering dominantly contributes to the omnipresent decay of electron diffraction spots in radiation-soft matter, which is why this approach can be exploited to stimulate time-temperature transformations that hinder/reduce electron-beam-induced object alterations during image acquisitions [17], and it even allows characterizing extended defects in radiation-sensitive, organic crystals at atomic resolution at room temperature without modifying their crystal structure [32].

## Figures and Tables

**Figure 1 nanomaterials-13-00971-f001:**
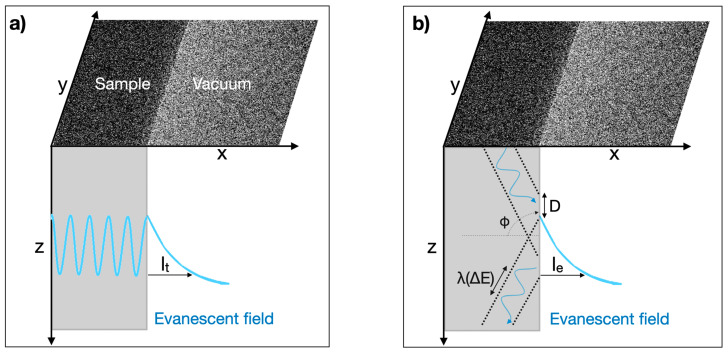
Schematic representation of evanescent field formations. (**a**) *Time-independent* solution: a trapped wave function tunnels into a potential barrier with an exponential decay length *l_t_*; (**b**) *Time-dependent* solution: an incident electron wave experiences total reflection and a Goos–Hänchen shift D that creates an evanescent field of exponential decay length *l_e_*.

**Figure 2 nanomaterials-13-00971-f002:**
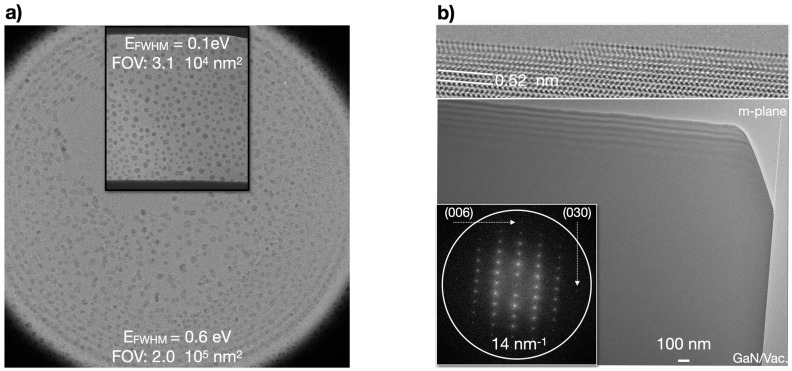
Microscope set-up and samples. (**a**) In a conventional microscope set-up, the depicted field of view (FOV) is limited by the aperture of the condenser system, which we used in combination with an electron beam of energy spread Δ*E_mc_* = 0.6 eV. The inset shows the parameters for a Nelsonian illumination scheme with Δ*E_mc_* = 0.1 eV. Calibration sample: gold on ultra-thin amorphous carbon. (**b**) A low-magnification image of the investigated GaN sample is shown in a-plane orientation together with a high-resolution image of the thin wedge at the top of the sample (basal plane) and its Fourier transform (inset). Contamination layers at surfaces were absent. In this work, the bright contrast surrounding the sample on a scale of 100 nm was investigated at the abrupt, nonpolar m-plane/vacuum interface.

**Figure 3 nanomaterials-13-00971-f003:**
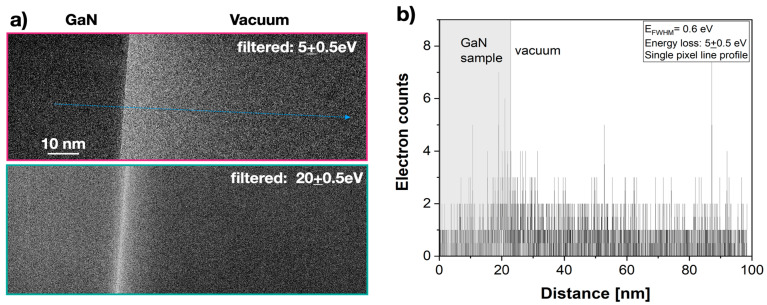
Amplitude image intensity from the sample/vacuum interface in the energy loss region. Fresnel oscillations were absent because Δf = 0. (**a**) Amplitude images with subtracted incident image intensity for 2 energy losses as indicated. Blue line: extraction of the line profile shown in (**b**). (**b**) Single-electron scattering events were detected with an outstanding S/N ratio. The resulting telegraph noise in the recorded EFTEM images was revealed by a line profile of single-pixel width. Averaging along the interface and the recording of image series were used for shot noise reduction.

**Figure 4 nanomaterials-13-00971-f004:**
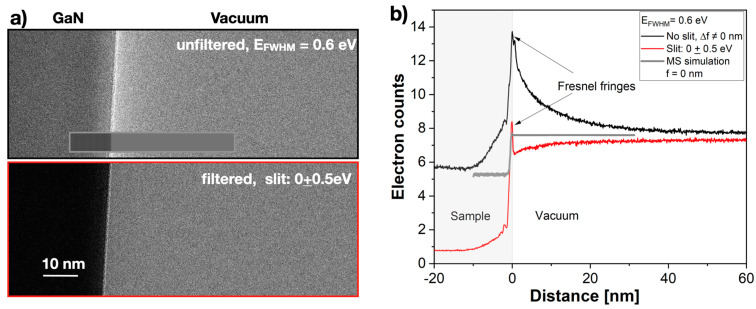
Intensity change by filtering the electron beam. The incident electron distribution was not subtracted. (**a**) Top: an as-recorded image intensity at the GaN/vacuum interface without energy filtering, and an MS simulation of an amplitude image inserted. Bottom: image intensity with an energy-filtered electron beam. (**b**) Averaged intensity profiles perpendicular to the interface. Fresnel fringes occurred if the defocus Δ*f* deviated from zero, and were suppressed in amplitude images, as shown by the MS simulation.

**Figure 5 nanomaterials-13-00971-f005:**
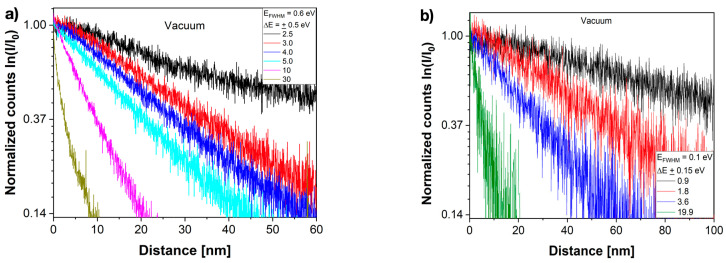
Selected intensity decays from the interface at *x* = 0 nm into the vacuum. Averaged data are normalized to 1 at the interface and plotted logarithmically. (**a**) Energy spread = 0.6 eV (**b**) Energy spread = 0.1 eV.

**Figure 6 nanomaterials-13-00971-f006:**
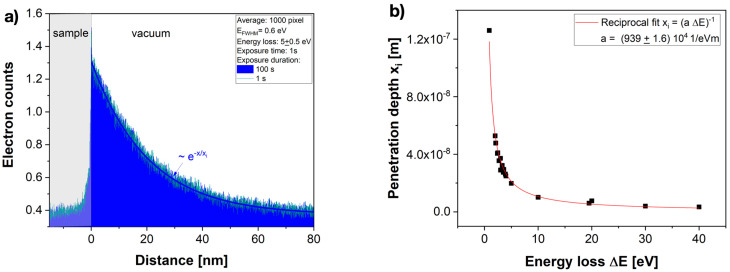
Energy-dependent penetration depth of the evanescent field. (**a**) The average electron count at the interface was about 1 electron. The intensity decay was strictly exponential *I*(0)*e*^−^*^x/xi^* with a characteristic distance x_i_ that we determined with a least square fit procedure for various energy losses to obtain the data set of Figure 6b. The effect was independent of the recording time. (**b**) A reciprocal dependence of *x_i_* on energy loss was revealed. In this case, a fitting parameter, a, could be determined and compared to the propagation of wave packets at light speed by the elemental constant ℏ*c* = 197 eVnm > 106 eVnm = ℏ*v* = 1/a.

**Figure 7 nanomaterials-13-00971-f007:**
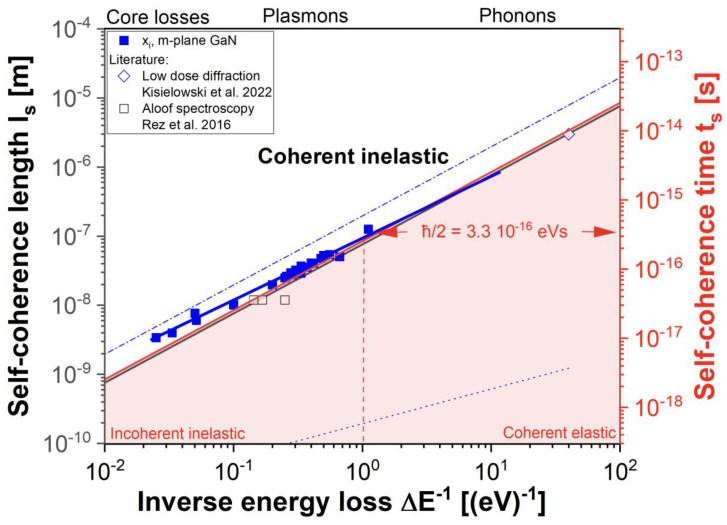
Spatial (black) and temporal (red) self-coherence versus inverse energy loss. Heisenberg’s uncertainty principle was revealed by choosing Δ*φ* = 0.5 rad for the self-interference of coexisting incident and scattered wave functions [12]. Measured penetration depth values *x_i_* (squares) and results from [17,27] (open symbols). They were compared to the self-coherence lengths *l_s_
*(black line), the tunneling distances *l_t_
*(dotted blue line), and the extension of evanescent fields l_e_ created by wave packets traveling at the speed of light, 197 eVnm = ℏ*c* (dash-dotted blue line), and at 54% of the speed of light, ℏ*v* = 106 eVnm (solid blue line). The abrupt coherence loss around 1 eV (dashed red line) separated modeling by either coherent–elastic or incoherent–inelastic scattering events in theoretical considerations [3,4,5]. Experimentally, no such difference seems present.

## Data Availability

Upon reasonable request, the utilized images can be obtained at cfkisielowski@sbcglobal.net.

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
