# Peer review of "Probing the Boundary between Classical and Quantum Mechanics by Analyzing the Energy Dependence of Single-Electron Scattering Events at the Nanoscale"

_nanomaterials, 2023, doi:10.3390/nano13060971_

Round 1

Reviewer 1 Report

In this paper the authors have experimentally and theoretically investigated electron self-coherence by monitoring the energy-dependent evanescent decay of the electron wave packet at the material/vacuum interface. The effect is associated with self-interference effects between the different electron scattered components due to coherent-inelastic Coulomb interactions.

The experiment is thoroughly conducted using a state-of-the-art aberration corrected transmission electron microscope. The data, together with the theoretical calculations, are original and of high quality. The article fits well within the scope of the journal, and the literature references are adequate to giving an overview of the recent advances in the field. The conclusions are physically sound and the quantification of the coherence length and time provides crucial information for coherent electron-matter interaction.

Therefore, I surely recommend the paper for publication.

So said, I believe that this paper would be of interest also to a wider portion of the physics community if the authors could formalize their theory in the context of a generic coherent-inelastic interaction. For instance, the quantification of the self-coherence length in the context of electron-light interaction (see for instance: https://doi.org/10.1063/5.0118096) would be a formidable result for the field of ultrafast electron microscopy, which is a new appealing recent trend in electron microscopy.

Reviewer 2 Report

The topic sketched by the title is of fundamental interest to a wide physics audience.
New demonstrations of peculiar effects from the "twilight zone" between the obviously classical (macroscopically intuitive) and the surely QM (microscopically "spooky") are most welcome.

line 50, classical

line 105, "in equation" to me sounds less confusing than "of equation"

The presentation is excellent in language and content. My own physics experience is in other fields than the TEM of microscopic samples discussed here, and I won't pretend that "all" is clear to me, considering the wide gaps in specific knowledge between the authors and me. However, the conceptual discussion makes sense to me, and the experiment seems very competently executed. The demonstrations are fascinating as well as illustrative.
I have no doubt that this manuscript is suitable for "nanomaterials" and should find worthy responses.

Reviewer 3 Report

The paper presents an outstanding experimental probe to establish the relation between the energy-dependent particle and wave descriptions of electron-matter interactions on the nanoscale. The presentation is excellent. The article should be accepted for publication in Nanomaterials.